# Stillbirth, neonatal and maternal mortality among caesarean births in Kenya and Uganda: a register-based prospective cohort study

Rakesh Ghosh ,[1] Nicole Santos,[1] Elizabeth Butrick,[2] Anthony Wanyoro,[3] Peter Waiswa,[4] Eliana Kim,[2] Dilys Walker[1,5]

RG and NS contributed equally.

[1]Institute for Global Health Sciences, University of California San Francisco, San Francisco, California, USA
[2]University of California San Francisco, San Francisco, California, USA
[3]Department of OB/GYN, Kenyatta University, Nairobi, Kenya
[4]School of Public Health, Makerere University, Kampala, Uganda
[5]University of California San Francisco Department of Obstetrics Gynecology and Reproductive Sciences, San Francisco, California, USA

**Correspondence to**
Dr Rakesh Ghosh;
Rakesh.Ghosh@ucsf.edu

## ABSTRACT

**Objective** To investigate the interaction of risks for adverse maternal and perinatal outcomes (stillbirth, predischarge neonatal and maternal mortality) among caesarean section (CS) compared with vaginal deliveries (VD).

**Design** Prospective cohort study.

**Setting** 10 CS-capable facilities in Busoga Region, East-Central Uganda and Migori County, Kenya.

**Participants** Individual birth data were extracted from maternity registers between October 2016 and April 2019. There were a total of 77 242 livebirths and 3734 stillbirths. Overall, 24% of deliveries were by CS with a range of 9%–49% across facilities.

**Primary outcome measures** Stillbirth, predischarge neonatal mortality and maternal mortality.

**Results** The adjusted ORs for stillbirth, predischarge neonatal mortality and maternal mortality after a CS were 1.3 (95% CI 1.1 to 1.6), 1.9 (95% CI 1.6 to 2.2) and 3.3 (95% CI 2.2 to 4.9), respectively, compared with a VD. The association between maternal mortality and CS was 3.9 (95% CI 2.8 to 5.5) when the delivery was a live birth and 1.7 (95% CI 1.0 to 3.0) when it was a stillbirth. Post hoc analyses showed that mothers who received a CS had a lower risk of stillbirth if they were documented as a referral.

**Conclusion** In this context, CS births were at higher risk for worse outcomes compared with VD. Better understanding of CS use and associated adverse outcomes within the mother–baby dyad is necessary to identify opportunities to improve quality of intrapartum care.

**Trial registration number** NCT03112018.

## Strengths and limitations of this study

⇒ This study uses data from over 80 000 births across 10 facilities in Kenya and Uganda with a range of caesarean section rates, to compare risk of intrapartum stillbirths, predischarge neonatal mortality, and facility-based pregnancy-related maternal death among women who underwent caesarean section compared to women who delivered vaginally.

⇒ This study explores risk of adverse outcomes among women who were referred in, and uncovers potential avenues of future exploration.

⇒ We lacked data on various maternal characteristics such as Robson criteria for caesarean section, antenatal care coverage, indication for caesarean section, perioperative and postoperative complications, and cause of death.

⇒ We also lacked data on system-related factors such as time between caesarean section decision and incision, access to anesthesia and personnel conducting the caesarean section.

## INTRODUCTION

As facility-based births increase, access to and performance of caesarean section (CS) is also increasing in many low-income and middle-income countries (LMICs). However, while global CS rates increased from 12% in 2000 to 21% in 2015, the increase has been only 5%–6% in East and South Africa.[1] CS rates are projected to near 30% globally by 2030, though sub-Saharan Africa is estimated to remain below 8%.[2]

Low CS rates may indicate unmet need and missed opportunity to prevent adverse outcomes.[3 4] Evidence from 16 sub-Saharan African countries between 2008 and 2011 showed that an average CS rate of approximately 4% was associated with increased maternal, neonatal and infant mortality, which may have been averted by increasing access to caesarean delivery for medically indicated cases.[5] Additionally, it is well documented that lack of trained personnel or access to safe anaesthesia, blood supply and essential drugs, contribute to CS-related adverse outcomes in many LMICs.[5–7] A review of 196 studies from 67 LMICs showed that rates of maternal death and stillbirth were 8 and 57 per 1000 CS procedures, and were highest in sub-Saharan Africa, at 11 and 83, respectively.[8] Given that CS is usually indicated for a maternal or neonatal complication, it is not surprising that CS is associated

with higher maternal and neonatal mortality. Access to timely and safe CS is critical to avert poor outcomes related to the mother–baby dyad.

Increasing CS rates to recommended levels, especially in resource-limited settings, may not always confer incremental benefit.[9–11] Overuse of CS in such settings introduce unnecessary risk for both mothers and newborns by diverting facility resources from the clinically indicated cases to low-risk pregnancies.[6 12]

While evidence separately links CS with maternal mortality and CS with neonatal outcomes in different settings, no data, to our knowledge, describes the interrelationship between maternal mortality and stillbirth among births by CS. Generally, if intrauterine demise is identified prior to delivery, there are few indications to perform a CS due to increased maternal risks.[13] To better understand this interaction, we investigated the characteristics including outcomes of CS performed across 10 facilities in Kenya and Uganda in a prospective birth cohort. In Uganda, facility-based CS rates were 22%–32% in general hospitals and 20%–25% in referral hospitals between 2012 and 2016.[14] In Kenya, 2014 Demographic and Health Survey data showed an overall 13% CS rate among institutional births, with notable disparities between rural and urban populations and across socioeconomic levels.[15] Data from the East Africa Preterm Birth Initiative (PTBi), a pair-matched cluster randomised trial (CRCT), provides a unique opportunity to investigate maternal and neonatal outcomes in the CS-capable facilities in the two countries.[16 17] In this study, we investigated the interaction of risks for adverse maternal and perinatal outcomes (stillbirth, predischarge neonatal and maternal mortality) among CS compared with vaginal deliveries (VD).

## METHODS

### Study design and setting

This analysis used maternity register data for deliveries between October 2016 and April 2019 from 10 CS-capable facilities in Busoga Region, East-Central Uganda and Migori County, Kenya that were involved in a pair-matched CRCT conducted by the East Africa PTBi.[16 17] Four facilities were private not-for-profit mission hospitals and six were public hospitals with 620–2420 and 1200–6700 deliveries per year, respectively. Four of the 10 facilities that are part of this study were in the control arm and three were in the intervention arm of the CRCT. The three remaining hospitals were large referral hospitals that received the intervention but were not part of the CRCT given the inability to pair match them.

Busoga Region and Migori County have a predominantly rural population of approximately three million and one million, respectively. Most women in the two respective regions deliver in a facility (77% and 53%), though regional neonatal mortality rates (27 and 19 per 1000 livebirths) and stillbirth rates (17 and 10 per 1000 pregnancies) remain high.[18 19] In Uganda, health centres

(level III) offer basic emergency obstetric care and are linked to a referring facility (either a health centre IV or a hospital) for comprehensive emergency obstetric care. In Kenya, referral to the county referral hospital from lower-level facilities is standard, though women may seek services within the private sector or across county borders.

### Data sources, eligibility criteria and variables of interest

Anonymised individual level birth data were extracted monthly from facility maternity registers by PTBi study staff. These routine data sources are completed by front-line healthcare workers. As part of the CRCT, all facilities received data strengthening, including reinforcement of key indicator definitions and continuous feedback on maternity register completion and accuracy for key variables (eg, gestational age, birth weight, Apgar, discharge status).[20 21] There was no routine follow-up after discharge.

Data collected include maternal (age, multiplicity, referral status) and newborn (birth weight, gestational age at birth, 5 min Apgar, infant sex) characteristics and maternal and newborn status at discharge. Maternal length of hospital stay was calculated using delivery and discharge dates. Mode of delivery was documented as CS, normal vaginal, assisted vaginal, breech and vacuum extraction. For this analysis, births were excluded if mode of delivery was missing, or documented as an abortion or vacuum aspiration. Babies born before arrival to the facility were also excluded.

The main outcomes were defined as: (1) stillbirths (birth at ≥28 weeks of gestation with no signs of life); (2) predischarge neonatal mortality (death of a liveborn baby before facility discharge) and (3) facility-based pregnancy-related maternal death.

Birth outcome was determined by triangulating data from delivery status, baby discharge status, and Apgar scores at 1 and 5 min. In both countries, stillbirths were categorised into intrapartum stillbirth (intrauterine death of a fetus during labour or delivery; recorded as 'fresh stillbirth') or antepartum stillbirth (intrauterine death of a fetus before the onset of labour, where the fetus showed degenerative changes, recorded as 'macerated stillbirth').[22] If either the delivery status or the baby discharge status were recorded as antepartum stillbirth or intrapartum stillbirth and the Apgar scores at 1 and 5 min were zero, their birth outcome was designated as documented. To reconcile the heterogeneity all stillbirths were categorised into a single group.

### Statistical analysis

The association of CS with maternal and perinatal outcomes was quantified using logistic regression and results are presented as ORs with 95% CIs. The data were two tiered—individual deliveries clustered within facilities. We used mixed effect models with a random intercept for each facility to account for the non-independence of the deliveries within facilities. When stratified by types of stillbirth, the number of outcomes reduced and the models did not converge. Hence, we fit single-level logistic

regression to gain insight into how the associations differ by stillbirth subtypes (supplemental results only).

To specify a parsimonious model, we used a directed acyclic graph (DAG)[23] and change-in-estimate confounder selection method.[24] The DAG identified a necessary and sufficient set of potential confounders that required adjustment. We then used the change-in-estimate method of statistical modelling to identify the final set of potential confounders, with 5% change in the exposure–outcome association as the criterion. More explanation about the statistical modelling and the directed paths are presented in online supplemental figure S1.

Interaction was investigated using a product term in the models. If the product term was significant at the 5% significance level, the models were stratified by the interacting variable. Led by the descriptive results on referrals, we framed a post hoc objective to examine interaction between CS and referral status on adverse outcomes. To minimise chances of false-positive associations, we adjusted for multiple comparisons using the method of false discovery rate.[25] The analysis was conducted in STATA V.16.1.

## Patient and public involvement

Our research question was informed by an observation that intrapartum stillbirth rates were high, especially among women who had CS. Since undergoing a CS involves added risk and a longer recovery period, we expect that women are more likely to consent to a CS if they anticipate a greater chance of an improved outcome. Given CS is often done for complications that lead to poorer outcomes, we decided to assess CS and maternal and perinatal outcomes in this context. We abstracted delivery and newborn care data for all women who delivered during the study period from the facility maternity register, under an IRB-approved waiver of consent. As this was a secondary analysis of a CRCT, all women who gave birth at a given facility, received the same standard of care. Patients were not involved in the design or conduct of the study, other than being a participant. Study results were shared with participating facilities' administration and staff by our in-country teams. However, results will not be disseminated directly to women.

## RESULTS

The final analytical dataset is shown in figure 1. PTBi EA collected information on 83 867 deliveries across the 10 CS-capable facilities, of which 80 976 had birth outcomes. To avoid double counting for multiple gestation pregnancies, we considered multiples as a single delivery event (n=78 451) for maternal outcomes.

There were 164 maternal deaths, 3734 stillbirths and 77 242 livebirths. The maternal mortality ratio was 212 per 100 000 livebirths; the stillbirth rate was 46 per 1000 births; and the predischarge neonatal mortality rate was 16 per 1000 livebirths.

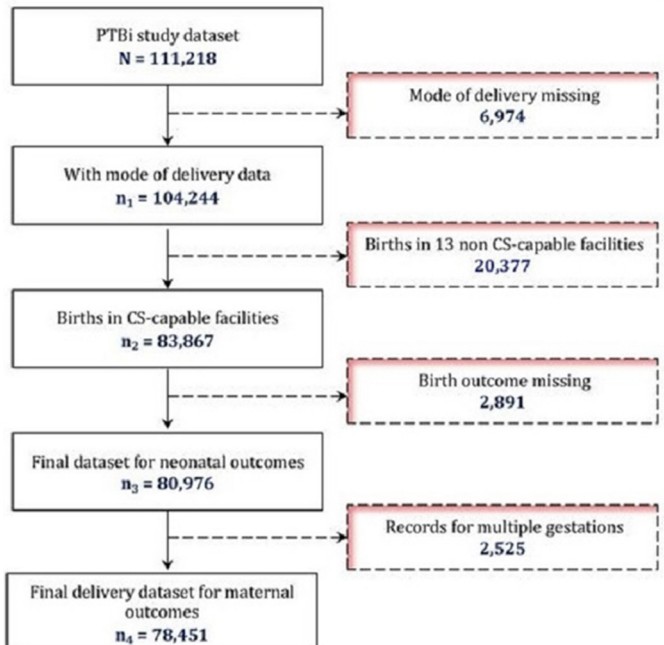

**Figure 1** Flow diagram describing the final analytical sample and exclusions. CS, caesarean section; PTBi, Preterm Birth Initiative.

Overall, 24% of deliveries were by CS (19 155/78 451) with a range of 9%–49% across facilities. Table 1 shows that private not-for-profit mission facilities performed more CS (38%) than public facilities (21%). Neither exposure to the PTBi intervention nor annual delivery volume were significantly associated with CS rates.

Maternal and neonatal characteristics by mode of delivery are shown in table 1. There was an increased association between CS and referred-in status, compared with those not referred. The length of hospital stay after delivery was significantly associated with mode of delivery with about two-thirds of mothers who were hospitalised for 7 days or longer having undergone a CS (OR 1.9, 95% CI 1.8 to 2.0). Overall, about 6% of women who underwent a CS had a hospital stay of more than 7 days compared with <1% for VD. The median (IQR) length of stay was 3 (1–5) days for a CS, compared with 1 (0–1) day for a VD. Over one-third of the women (36%) who had multiple gestations underwent CS, with an OR of 1.8 (95% CI 1.5 to 2.0), relative to singletons.

For neonatal characteristics, relative to infants with gestational age (GA) between 37 and 40 weeks, those who were <32 completed weeks at birth had reduced odds for CS, while those between 32 and <37 weeks and >40 weeks had increased odds for CS (table 1). Likewise, relative to infants between 2500 and 3999 g, those weighing <1000 g had reduced odds of CS, while those between 1000 and 2499 g and those ≥4000 g had increased odds of CS. There was an increased association between CS and Apgar <7 at 5 min (OR 1.8, 95% CI 1.4 to 2.5), compared with those with an Apgar score ≥7.

Table 2 presents the adjusted OR (aOR) for the association between CS and adverse maternal or perinatal

**Table 1** Facility, maternal and neonatal characteristics for vaginal deliveries and caesarean sections

| | Vaginal delivery % (n) | Caesarean section % (n) | Odds ratio (95% CI) | P value (trend) |
|---|---|---|---|---|
| Facility characteristics | | | | |
| Facility type* | | | | |
| Public | 78.7 (50 049) | 21.3 (13 572) | 1.00 | <0.001 |
| Private | 62.4 (9247) | 37.7 (5583) | 3.07 (1.71 to 5.51) | |
| Exposure to PTBi intervention* | | | | |
| No | 75.7 (50 378) | 24.4 (16 218) | 1.00 | 0.455 |
| Yes | 75.3 (8918) | 24.8 (2937) | 1.51 (0.51 to 4.46) | |
| Annual delivery volume* | | | | |
| <1000 (low) | 68.4 (918) | 31.7 (425) | 1.46 (0.65 to 3.28) | 0.594 |
| 1000 to <2800 (medium) | 73.0 (16 890) | 27.0 (6251) | 1.00 | |
| ≥2800 (high) | 76.9 (41 488) | 23.1 (12 479) | 0.76 (0.28 to 2.04) | |
| Maternal and neonatal characteristics | | | | |
| Maternal age (years)* | | | | |
| <18 | 76.4 (3788) | 23.6 (1168) | 1.06 (0.90 to 1.26) | 0.823 |
| 18 to ≤35 | 75.7 (51 924) | 24.3 (16 686) | 1.00 | |
| 36 to 53 | 74.0 (3151) | 26.1 (1110) | 1.04 (0.93 to 1.16) | |
| Delivery referred in* | | | | |
| No | 75.5 (32 823) | 24.5 (10 661) | 1.00 | 0.17 |
| Yes | 55.0 (3847) | 45.0 (3146) | 2.33 (2.06 to 2.64) | |
| Missing | 80.9 (22 542) | 19.1 (5324) | 0.98 (0.83 to 1.15) | |
| Day of delivery* | | | | |
| Monday–Friday | 75.2 (41 781) | 24.7 (13 773) | 1.00 | 0.009 |
| Weekend | 76.7 (16 735) | 23.3 (5086) | 0.93 (0.88 to 0.98) | |
| Maternal length of stay* | | | | |
| <7 days | 76.7 (57 593) | 23.3 (17 462) | 1.00 | <0.001 |
| ≥7 days | 33.4 (543) | 66.6 (1083) | 1.93 (1.82 to 2.04) | |
| Multiple gestation* | | | | |
| Singleton | 75.9 (57 717) | 24.1 (18 286) | 1.00 | <0.001 |
| Twins, triplets or quadruplets | 64.5 (1579) | 35.5 (869) | 1.77 (1.53 to 2.04) | |
| Preterm birth (<37 completes weeks)* | | | | |
| No | 75.9 (52 493) | 24.1 (16 708) | 1.00 | 0.016 |
| Yes | 73.6 (6803) | 26.5 (2447) | 1.08 (1.01 to 1.15) | |
| Gestational age (weeks)* | | | | |
| 20 to <28 | 85.8 (338) | 14.2 (56) | 0.57 (0.41 to 0.80) | <0.001 |
| 28 to <32 | 80.8 (1337) | 19.2 (318) | 0.74 (0.60 to 0.91) | |
| 32 to <37 | 74.1 (5170) | 25.9 (1809) | 1.14 (1.03 to 1.25) | |
| 37 to <40 | 75.7 (34 303) | 24.3 (10 986) | 1.00 | |
| ≥40 | 76.2 (8811) | 23.8 (2754) | 1.08 (1.01 to 1.15) | |
| Low birth weight (<2500 g)† | | | | |
| No | 75.9 (52 819) | 24.1 (16 777) | 1.00 | <0.001 |
| Yes | 71.1 (5926) | 28.9 (2405) | 1.22 (1.12 to 1.33) | |
| Birth weight (g)† | | | | |

Continued

**Table 1** Continued

|  | Vaginal delivery % (n) | Caesarean section % (n) | Odds ratio (95% CI) | P value (trend) |
|---|---|---|---|---|
| <1000 | 89.9 (257) | 10.1 (29) | 0.34 (0.18 to 0.66) | <0.001 |
| 1000–2499 | 71.1 (5926) | 28.9 (2405) | 1.24 (1.14 to 1.35) | |
| 2500–3999 | 76.2 (50 006) | 23.8 (15 599) | 1.00 | |
| ≥4000 | 70.5 (2813) | 29.5 (1178) | 1.39 (1.25 to 1.55) | |
| Apgar score at 5 min† | | | | |
| ≥7 | 75.7 (47 154) | 24.3 (15 159) | 1.00 | <0.001 |
| <7 | 63.0 (2135) | 37.0 (1255) | 1.78 (1.31 to 2.40) | |

*Based on the delivery dataset of 78 451 mothers.
†Based on the neonatal dataset of 80 976.
PTBi, Preterm Birth Initiative.

health outcomes. The aORs were 1.3 (95% CI 1.1 to 1.6) for stillbirth and 1.9 (95% CI 1.6 to 2.2) for predischarge neonatal mortality when the mode of delivery was CS compared with VD. The aOR for maternal death after a CS was 3.3 times higher (95% CI 2.2 to 4.9) compared with VD. Online supplemental table S1 presents CS associations by stillbirth types, which is decreased for antepartum stillbirth (0.7, 95% CI 0.7 to 0.8) and increased for intrapartum stillbirth (1.8, 95% CI 1.6 to 2.0).

Figure 2 shows that the risk for maternal death after CS was higher for a livebirth than for a stillbirth. For livebirths, the aOR for maternal death after CS was 3.9 (95% CI 2.8 to 5.5), compared with VD, while it was 1.7 (95% CI 1.0 to 3.0) for stillbirths. For maternal mortality after CS, the association did not statistically differ by referral status of the mothers. Numerical estimates are presented in online supplemental table S2.

Figure 3 shows that when the mother was not referred, the aOR for stillbirth after a CS was 1.2 (95% CI 1.0 to 1.5), compared with a VD and 0.9 (95% CI 0.8 to 1.0)

when the delivery was referred-in. For predischarge neonatal mortality, the association did not statistically differ by referral status of the mothers. Online supplemental table S3 presents numerical estimates for figure 3.

Apgar score <7 at 5 min is an indicator of neonatal well-being, as well as success of resuscitation. Online supplemental table S3 shows that when the mother was not referred, the odds for Apgar<7 after CS was 2.0 (95% CI 1.6 to 2.5), compared with a VD and 1.2 (95% CI 1.1 to 1.5) when the delivery was referred-in.

The statistical significance for the associations presented in table 2 as well as the interactions remained largely unchanged after adjustment for multiple comparisons (online supplemental tables S2 and S3).

## DISCUSSION

This analysis demonstrates an increased association of CS with adverse maternal and perinatal outcomes in two sub-Saharan countries. Compared with women who had

**Table 2** Adjusted* ORs for neonatal and maternal outcomes among those who had caesarean sections

|  | Vaginal delivery % (n) | Caesarean section % (n) | Adjusted OR (95% CI) | P value | Adjusted p value† |
|---|---|---|---|---|---|
| Stillbirth‡ | | | | | |
| No | 75.5 (58,338) | 24.5 (18 904) | 1.00 | 0.006 | 0.01 |
| Yes | 69.0 (2,576) | 31.0 (1158) | 1.33 (1.09 to 1.63) | | |
| Predischarge neonatal mortality‡ | | | | | |
| No | 75.4 (60 174) | 24.6 (19 606) | 1.00 | <0.001 | 0.0003 |
| Yes | 61.9 (740) | 38.1 (456) | 1.88 (1.64 to 2.16) | | |
| Maternal mortality§ | | | | | |
| No | 75.2 (52,694) | 24.8 (17 419) | 1.00 | <0.001 | 0.0003 |
| Yes | 49.4 (77) | 50.6 (79) | 3.28 (2.21 to 4.88) | | |

*Adjusted for birth weight, annual delivery volume, type of facility and country and exposure to Preterm Birth Initiative intervention.
†P values adjusted for multiple comparison using false discovery rate method.
‡Based on the neonatal dataset of 80 976.
§Based on the delivery dataset of 78 451 mothers.
PTBi, preterm birth initiative.

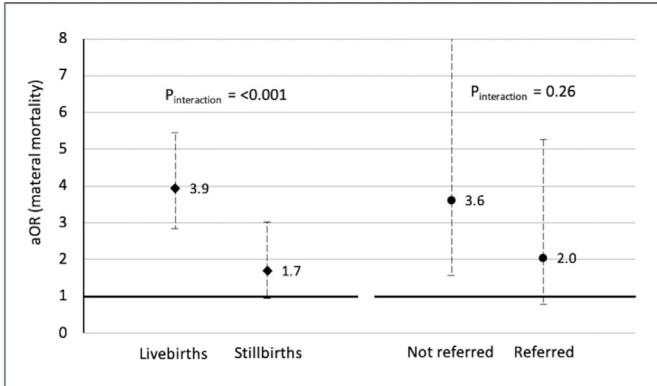

**Figure 2** Adjusted! OR and 95% CI for facility-based pregnancy related death and caesarean section stratified by birth outcome and referral status[!]. [!] Adjusted for birth weight, annual delivery volume, type of facility, country and exposure to PTBi intervention. Note—After adjusting for multiple comparison using false discovery rate method, the interaction p values changed to 0.0003 and 0.29, respectively. PTBi, Preterm Birth Initiative.

a VD, those who underwent CS had increased association with maternal death, stillbirth and predischarge neonatal mortality. We found a higher association between CS and maternal mortality for a livebirth than for a stillbirth. This novel differential association by birth outcome is subject to independent verification but underscores the need to approach the clinical situation and the decision-making from the lens of the mother–baby dyad that optimises improved outcomes. We also found that women who were documented as referred-in and underwent a CS had a lower risk of stillbirth than those who were not referred-in and underwent a CS. While the risk for maternal mortality after a CS was not statistically different by referral status, it generally trended towards lower risk of poor outcome if documented as referred-in.

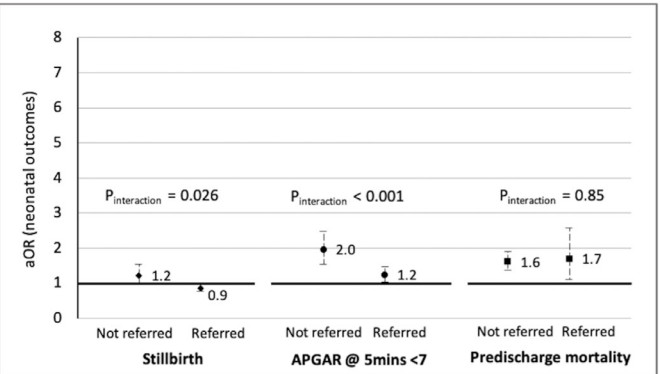

**Figure 3** Adjusted[!] OR and 95% CI for adverse neonatal outcomes and caesarean section stratified by referral status[!]. [!] Adjusted for birth weight, annual delivery volume, type of facility, country and exposure to PTBi intervention. Note—After adjusting for multiple comparison using false discovery rate method, the interaction p values changed to 0.04 for stillbirth, to 0.003 for Apgar, and remained unchanged at 0.85 for predischarge mortality. PTBi, preterm birth initiative.

Our overall finding that women who had a CS had higher risk of maternal death is consistent with previous studies and not surprising given the indications for CS include high-risk conditions for both mothers and newborns. In a South African study, CS rates at various facilities ranged between 15% and 28% and the risk of maternal death after CS was 2.8 times that of VD.[26] A study across 41 referral hospitals in Mali and Senegal demonstrated that women who underwent intrapartum CS were at higher risk for maternal mortality compared with women who had a spontaneous VD, as well as predischarge neonatal mortality.[27]

Our findings suggest the risk of maternal death after a CS was higher for a livebirth than it was for a stillbirth. With limited relevant data in the registries and no access to maternal death reviews, we could not ascertain the cause and time of death, associated complications, and time between decision to incision, limiting the ability to interpret these results. At Mulago National Referral Hospital in Uganda, shorter decision-to-CS delivery times were associated with fetal distress, while longer times were associated with pre-eclampsia and premature rupture of membranes (PROM), hinting towards increased attention and action to ensure fetal well-being.[28] This could be because mothers who have pre-eclampsia or PROM may require stabilisation or antenatal corticosteroids prior to CS, unlike mothers who are diagnosed with non-reassuring fetal heart tones. However, elevated odds of maternal death even with stillbirth (but less than live birth) bring to light the need for careful evaluation on indications for caesarean and quality of caesarean care.

We found higher odds of stillbirth among women who received a CS compared with VD. Studies have linked unmet need for CS to increased intrapartum stillbirths.[4 29] In this study, more than 30% of stillbirths were delivered by CS suggesting that intrapartum quality of care and CS timeliness may contribute to stillbirth outcomes. Nonetheless, it raises questions of adequate fetal monitoring prior to CS initiation, appropriate indication for CS, and timeliness. It is unclear whether these were conducted for clinically appropriate indications, such as uterine rupture, prior CS, or if there was a lack of alternative capacity to manage stillbirth. This paucity of data highlights the value of auditing CS cases particularly those in which a stillbirth is delivered. This may reveal important and urgent opportunities for quality improvement of intrapartum care in these settings.

Increased CS rates among referred-in mothers could contribute to higher rates of stillbirths[30 31] and maternal deaths,[26] due to delays in transportation, timeliness, etc. However, we were intrigued to note that referred-in mothers had a lower risk of stillbirth and maternal death, though the latter was not significant. We also found that compromised newborn well-being at birth (Apgar score <7 at 5 min) showed a similar relationship, suggesting that referred-in deliveries may have been exposed to more timely intervention or increased evidence-based practices during the intrapartum period,

such as newborn resuscitation. While the underlying contributors to this finding are unclear; some hypotheses might be: (1) referred-in women were stabilised prior to arrival; (2) prior assessment led to more timely CS on arrival at the receiving facility; (3) referred-in women had more time to mobilise resources for a timely CS and (4) decision to refer was made before the onset of severe complications (eg, early diagnoses of CS indication) or due to non-medical reasons. Missingness of the referral variable demands cautious interpretation. However, the intriguing results suggest that the existing referral system may be functional in identifying and managing high-risk women and their newborns. At the least, this finding deserves further exploration in this particular referral system and in similar contexts for a deeper understanding.

Collectively, these findings highlight the need to assess gaps in CS quality of care and system bottlenecks to improve maternal and perinatal outcomes. CS-specific checklists and audits have improved uptake of evidence-based practices, clinical documentation and quality of care.[32][33] Few vacuum or forceps deliveries were recorded in maternity registers, which should be optimised to avoid unnecessary CS, particularly among known intrauterine fetal deaths.[34][35] Another obvious area of improvement is documentation of clinical diagnoses, indications for CS, and referral status. In the absence of robust data for clinical indications for CS, we cannot rule out elective CS, though clinically indicated CS likely constitute a vast majority of all CS in this context.[36] While PTBi EA invested in data strengthening across the study facilities, this largely focused on variables such as gestational age, birth weight, Apgar and discharge status.[16] Maternity registers use free text to document final diagnosis, delivery complications and referral status. In this study, among women who had CS and stillbirth, 40% had no diagnosis documented; 63% and 43% of women who died after a VD or a CS had no diagnosis, respectively. Due to these data issues, we cannot assess whether conditions such as preoperative or perioperative obstetric haemorrhage, hypertensive disorders, infection or anaesthesia complications may have contributed to deaths, all of which have been independently associated with maternal mortality.[37] Given the relatively high CS rates in some of the facilities and the rising rates globally, inclusion of CS indications may be warranted in birth registries.

Likewise, differentiation between intrapartum and antepartum stillbirths would undoubtedly inform opportunities and curb the potential overuse of CS for non-indications, particularly in cases of known stillbirth. Overall, rate of intrapartum stillbirths has previously been suggested as a maternal quality care indicator by WHO and other stakeholders.[38] Disaggregating this indicator by mode of delivery could serve as a quality of intrapartum care indicator, particularly in the context of rising CS rates across LMICs[39] and the fact that CS should be considered as a last resort for delivering a stillborn infant due to the increased risk to the mother.[13]

## Strengths and limitations

The study has several limitations. Outcomes were ascertained by facility personnel and there may be data quality issues as maternity registers are often completed in the context of high workload. For example, approximately 40 cases had conflicting delivery and discharge status, and were excluded. Second, we lacked robust data on various maternal characteristics such as Robson criteria for CS, antenatal care coverage, indication for CS, perioperative and postoperative complications, and cause of death. We also lacked data on system-related factors such as time between CS decision and incision, access to anaesthesia, and personnel conducting the CS. As noted above, this information would have helped contextualise our findings. CS and VD are very different by its very nature leaving little room for misclassification. Grouping all stillbirths into one, further reduces possibility of misclassification by type. However, there is room for potential misclassification between stillbirths and predischarge neonatal mortality, but such misclassification is unlikely to happen among the CS, as these are conducted by specialists.

Regardless of these limitations, we used robust design and analytical approaches to quantify associations and have been cautious in drawing conclusions. By leveraging existing data systems, we captured real-world contexts and underscore the importance of data standardisation, completeness and accuracy. Another strength of this study was the availability of pregnancy-related mortality data across heterogeneous types of facilities in two countries. Given the duration of the PTBi EA programme, we had sufficient sample size to power the investigation for maternal mortality and investigate a novel interaction between CS and birth outcome. This study provides rare evidence related to outcomes of CS from a low-resource setting and the results may be generalisable to similar contexts and settings. The results are unlikely to be chance findings, given the significance level after adjustment for multiple comparisons.

## CONCLUSION

In this study, CS births were at higher risk for worse maternal and perinatal outcomes compared with VD. Additionally, the risk of maternal death after a CS was higher among livebirths than it was for stillbirths. Better understanding of CS use and associated adverse outcomes within the mother–baby dyad is necessary to identify opportunities to improve quality of intrapartum care. Detailed case documentation including indications for CS, referral and causes of death is needed to shed light on these issues and identify pathways to improve care.

**Correction notice** This article has been corrected since it was published Online First. The liecence has been updated to CC BY.

**Acknowledgements** We appreciate all the facility healthcare workers, leadership and administration who made this work possible. We thank all members of the Preterm Birth Initiative Kenya and Uganda Implementation Research Collaborative, particularly the data teams who helped strengthen and collect data from facility maternity registers. We thank Phelgona Otieno who was the Kenya PI for the parent

trial. In particular, we convey our heartfelt thanks to the study participants for their participation and immensely valuable contribution to this study.

**Contributors** DW, EB, NS and RG contributed to study conceptualisation. RG and EK helped in data cleaning and preparing the data for analysis. RG led data analysis. RG and NS drafted and revised the manuscript. DW, PW, AW, EK, EB, NS and RG contributed to data synthesis and interpretation, review and approval of the final version of the manuscript for submission. DW and PW as PIs for the parent trial were responsible for overall design, implementation and data collection.

**Funding** This work was supported by the East Africa Preterm Birth Initiative, which was funded by the Bill and Melinda Gates Foundation (OPP1107312).

**Competing interests** None declared.

**Patient and public involvement** Patients and/or the public were not involved in the design, or conduct, or reporting, or dissemination plans of this research.

**Patient consent for publication** Not applicable.

**Ethics approval** The PTBi EA CRCT was granted ethical approvals from Higher Degrees, Research and Ethics Committee (MUSPH HDREC 395) from Makerere University; KEMRI Scientific and Ethics Review Committee (KEMRI/ SERU/CCR/0034/3251) and UCSF Committee on Human Research (16–19162). Permission to extract non-identifiable data from maternity registers was allowed under these approvals.

**Provenance and peer review** Not commissioned; externally peer reviewed.

**Data availability statement** All data relevant to the study are included in the article or uploaded as online supplemental information. Anonymised data will be made available on reasonable request to: Dilys.Walker@ucsf.edu.

**ORCID iD**
Rakesh Ghosh http://orcid.org/0000-0002-7839-4148

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
