## [Reviewer comments · BMJ Open]

ARTICLE DETAILS

TITLE (PROVISIONAL)	Stillbirth, neonatal, and maternal mortality among cesarean births in Kenya and Uganda: a register-based prospective cohort study
AUTHORS	Ghosh, Rakesh; Santos, Nicole; Butrick, Elizabeth; Wanyoro, Anthony; Waiswa, Peter; Kim, Eliana; Walker, Dilys

VERSION 1 – REVIEW

REVIEWER	Agrawal, Swati Lady Hardinge Medical College, Obs & Gynae
REVIEW RETURNED	24-Oct-2021

GENERAL COMMENTS	The study is an exhaustive one and addresses a pertinent issue which should be publication.
---

REVIEWER	Dhaded, Sangappa Jawaharlal Nehru Medical College, Women's and Children's Health
REVIEW RETURNED	27-Dec-2021

GENERAL COMMENTS	I congratulate the author for focusing on a very important issue related to maternal and newborn health in developing world. I have few suggestions for minor revisions. 1. Line 59: To replace "within the" by "related to"2. Line 86: To delete 'people'3. Line 88: To replace "pregnancies" by "births"4. Line 96: To replace "of" by "on"5. Line 151 to 155 should be included in acknowledgement.6. Line 226: To replace "a mother who is" by "mothers who are"7. Page 26, Figure 1 should be changed to Figure 2.
--

VERSION 1 – AUTHOR RESPONSE

REVIEWER #1 COMMENTS

Dr. Swati Agrawal, Lady Hardinge Medical College

The study is an exhaustive one and addresses a pertinent issue which should be publication.

We appreciate the positive feedback.

REVIEWER #2 COMMENTS

Dr. Sangappa Dhaded, Jawaharlal Nehru Medical College

I congratulate the author for focusing on a very important issue related to maternal and newborn health in developing world. I have few suggestions for minor revisions.

We appreciate the positive feedback for this reviewer. We have made the revisions below.

1. Line 59: To replace "within the" by "related to" – revised.
2. Line 86: To delete 'people' – revised.
3. Line 88: To replace "pregnancies" by "births" – These data referring to stillbirths is defined in the countries' Demographic and Health Survey (DHS) data as fetal deaths in pregnancies lasting seven or more months. The metric used in these citations are per 1,000 pregnancies, so we have not changed the denominator to births.
4. Line 96: To replace "of" by "on" - revised.
5. Line 151 to 155 should be included in acknowledgement. This paragraph was duplicated in the Acknowledgement section. We have removed it from the Patient and Public Involvement section.
6. Line 226: To replace "a mother who is" by "mothers who are" - revised.
7. Page 26, Figure 1 should be changed to Figure 2. - revised.

OTHER CORRECTION TO MANUSCRIPT: In reviewing our manuscript to respond to reviewers' comments, we identified another Uganda maternity register variable that allowed us to differentiate between antepartum and intrapartum stillbirths (i.e., recorded as macerated versus fresh). All stillbirths included in this study were categorized as one of these two types of stillbirths. Therefore, we have updated the manuscript text to reflect this modification, as well as Supplemental Table S1 whereby undefined stillbirths have been re-categorized. We note that this revision neither changes the main results nor the conclusions because they are based on overall stillbirths (not by types). The revised results remain consistent with our prior results – specifically, decreased C-section odds if the fetus was identified as an antepartum stillbirth and an increased C-section odds if the fetus was identified as an intrapartum stillbirth. This result is consistent with what you would expect clinically.